# The Study of Radioactive Contaminations within the Production Processes of Metal Titanium for Low-Background Experiments

**DOI:** 10.3390/ma17040832

**Published:** 2024-02-09

**Authors:** Marina Zykova, Elena Voronina, Alexander Chepurnov, Mikhail Leder, Maria Kornilova, Alexey Tankeev, Sergey Vlasov, Alexander Chub, Albert Gangapshev, Ali Gezhaev, Dzhamilya Tekueva, Igor Avetisov

**Affiliations:** 1Department of Chemistry and Technology of Crystals, Mendeleev University of Chemical Technology, 125047 Moscow, Russia; zykova.m.p@muctr.ru (M.Z.); voronina.e.v@muctr.ru (E.V.); 2Skobeltsyn Institute of Nuclear Physics, Lomonosov Moscow State University, 119234 Moscow, Russia; 3Radiation Physics Laboratory, Belgorod State National Research University, 308015 Belgorod, Russia; 4PJSC “VSMPO-AVISMA Corporation”, 624760 Verkhnyaya Salda, Russia; leder_mo@vsmpo-avisma.ru (M.L.); kornilova_ma@vsmpo-avisma.ru (M.K.); tankeev1972@avisma.ru (A.T.); vlasovsa@avisma.ru (S.V.); 5Modern Technologies Ltd., 618541 Solikamsk, Russia; chub51207@mail.ru; 6Baksan Neutrino Observatory, The Institute for Nuclear Research of the Russian Academy of Science, 117312 Moscow, Russia; gangapsh@inr.ru (A.G.); gezhaev@gmail.com (A.G.); t.jami.a@mail.ru (D.T.)

**Keywords:** titanium sponge, structural titanium, pure substance, mass spectrometry with inductively coupled plasma, gamma spectroscopy with high-purity germanium detectors, radionuclides

## Abstract

Ultra-low-radioactivity titanium alloys are promising materials for the manufacture of low-background detectors which are being developed for experiments in astroparticle physics and neutrino astrophysics. Structural titanium is manufactured on an industrial scale from titanium sponge. The ultra-low-background titanium sponge can be produced on an industrial scale with a contamination level of less than 1 mBq/kg of uranium and thorium isotopes. The pathways of contaminants during the industrial production of structural titanium were analyzed. The measurements were carried out using two methods: inductively coupled plasma mass spectroscopy (ICP-MS) and gamma spectroscopy using high-purity germanium detectors (HPGes). It was shown that the level of contamination with radioactive impurities does not increase during the remelting of titanium sponge and mechanical processing. We examined titanium alloy samples obtained at different stages of titanium production, namely an electrode compaction, a vacuum arc remelting with a consumable electrode, and a cold rolling of titanium sheets. We found out that all doped samples that were studied would be a source of uranium and thorium contamination in the final titanium alloys. It has been established that the only product allowed obtaining ultra-low-background titanium was the commercial VT1-00 alloy, which is manufactured without master alloys addition. The master alloys in the titanium production process were found cause U/Th contamination.

## 1. Introduction

Low-background experiments in the field of particle physics represent an actively developing area of modern experimental physics. The goal of these experiments is to answer a number of fundamental questions about the present structure of the Universe, the nature of dark matter particles [1,2], and the fundamental properties of neutrinos [3]. Low-background experiments are carried out in underground low-background laboratories [4,5]. Due to the fact that most of these experiments need a target operating at cryogenic temperatures, the main structural element of corresponding detectors is a low-background cryostat. The main trend in the development of modern low-background detectors is an increase in their sensitive volumes due to an increase in the mass of the detector’s construction elements. This leads to the fact that the size of cryostats increases significantly and the amount of required low-background structural materials increases to tens of tons. The previously used approach of selecting material with the required low radioactivity from a warehouse does not work for such masses of structural materials. It is necessary to develop industrial methods for the production of low-background materials.

Promising low-background materials are titanium alloys. Titanium sponge is the primary product for all titanium alloys. It is produced from simple chemical compounds—metal magnesium and titanium tetrachloride (TiCl_4_)—using the magnesium thermal reduction method [6]. It is obvious that if the initial products have ultra-low-background properties, then the products obtained from them must also have ultra-low-background properties if the ingress of radioactive impurities is avoided during the manufacturing process. Our previous studies have shown that TiCl_4_, obtained directly and locally within the production cycle of titanium sponge via the deep distillation of chlorides of ore concentrates, has the necessary low-background properties [7], whereas metal magnesium requires special preparation. Commercial metal magnesium is a proven source of radioactive impurities in titanium sponge [6]. However, it has been shown that titanium sponge can be produced with ultra-low levels of radioactive impurities, below 1 mBq/kg.

In order to prove the possibility of manufacturing structural titanium alloys, it is necessary to analyze the stages of titanium sponge transformation into a titanium alloy and study the migration of radioactive impurities in industrial processes accompanying these stages. For this purpose, samples of materials were selected from various stages of titanium alloy production. Quantitative analysis was carried out using the ICP-MS method for the total content of uranium and thorium isotopes. Massive samples of titanium sponge and metallic titanium were additionally analyzed by gamma spectroscopy with high-purity germanium detectors located in the underground laboratory.

## 2. Materials and Methods

### 2.1. Methods for Studying Samples Using ICP-MS

Inductively coupled plasma mass spectrometry was used to analyze the chemical purity of all samples. The samples were transferred from the solid sample to the liquid phase. The mass of the sample was at least 1 g. Alloy samples were dissolved in 20 mL of hydrochloric acid (7N) purified by surface distillation systems (BSB-929-IR, BERGHOFF GmbH&Co., Wenden, Germany) in polytetrafluoroethylene autoclaves (DAP-100, PTFE, BERGHOFF GmbH&Co., Wenden, Germany) using a SPEEDWAVEFOUR microwave decomposition setup (BERGHOFF GmbH&Co., Wenden, Germany). The dissolution product was transferred into a polypropylene (PP) tube. Then, 1 mL of aliquot was diluted to 15 mL. Deionized water was obtained using an Aqua-MAX-Ultra 370 Series unit (Young Lin Instruments Co., Ltd., Seoul, Republic of Korea) and had an electrical resistance of 18.2 MΩ cm. Then, the resulting solution was transferred to a polypropylene test tube, and then the solution was diluted with water. The prepared solution was analyzed using inductively coupled plasma mass spectrometry (ICP-MS). Analytical measurements were carried out using an inductively coupled plasma mass spectrometer NexION300D (PerkinElmer Inc., Waltham, MA, USA). The quantitative analysis of Th and U was carried out using the “additive” method, taking into account the concentration of the main (matrix Ti) elements in the analyzed solution. Standard solutions (PerkinElmer Inc. Waltham, MA, USA) were used for calibration. Measurements were carried out using isotopes of ^232^Th and ^238^U.

The optimized operating mode of the NexION300D spectrometer for the impurity analysis of samples with Ti matrix element is presented in Table 1.

### 2.2. Methodology for Analyzing Radioactive Impurities in Titanium Metal Samples at Various Stages of Production Using the HP-Ge Method

#### 2.2.1. Description of the Installation and Measurement Technique

The measurements were carried out on low-background gamma spectrometers made of ultra-pure germanium, located inside a passive shield consisting of ~20 cm of copper, ~15 cm of lead, and ~8 cm of borated polyethylene. The installation is located in the underground laboratory of the Baksan Neutrino Observatory (BNO INR RAS). The detector’s sensitive element had a diameter of 60.6 mm and a height of 34.2 mm and was packaged in a thin-walled aluminum housing made of ultra-low-background aluminum with a wall thickness of 1.5 mm (Figure 1a).

Bulk products (titanium sponge) were placed in low-background plastic conical containers made of polyethylene (an upper diameter of 90 mm, a lower diameter of 80 mm, a height of 45 mm at the top of the lid, and a height of 40 mm inside) with a volume of 180 mL and a weight of 13 g. The samples were washed with alcohol to remove dust before placing them into containers. The container with the sample was placed on the upper edge of the detection unit casing.

Round bars in a U-shaped section were machined from titanium ingots, which were “put on” the detection unit (Figure 1b).

Disks and rings were machined from titanium sheets to give the U-shape structure the same shape as the U-shaped bars made of titanium ingots. The same sample shape for sheets and ingots simplified the process of comparing spectrometric measurement results.

The samples of titanium ingots and sheets were etched with a solution of 20% nitric acid and 1% hydrofluoric acid, before being washed in high-purity low-radioactivity deionized water.

The relative efficiency of the HPGe for the applied geometries of the samples was 20%.

#### 2.2.2. Calculation Technique

The aim of measuring the gamma radiation spectra of the samples using the HP-Ge method is to determine its specific residual radioactivity, which allows the volumetric concentration of radioactive isotopes to be restored at the levels of mBq units and below. This mathematical task belongs to the class of inverse problems and its solution is only possible with a sufficient amount of a priori data on the object of the study. The correct reconstruction of radiation sources from measured spectra is only possible if the propagation of gamma radiation in the detector and the sample itself is correctly taken into account. To do this, it is necessary to simulate the propagation of the detector’s own gamma background together with the expected background from the sample.

Calculations on the efficiency of the registration of γ-quanta from radionuclide decays in the samples under study were made using the MCC-MT (Monte Carlo Calculation Multi Thread) software package [8] included with low-background gamma spectrometers. This program allows a 3D model of the installation, including the detector, the detector’s structural elements, and the sample, to be built, and allows a predefined set of radionuclide decays to be simulated in the sample. The output of the simulation is the spectrum of energy release in the detector. The number of events at the peak total absorption of gamma rays was determined from the model spectra. The branching factors in decay chains and the quanta emission per decay of nuclide were taken into account. The results of the calculation based on the simulation are presented in Table 2.

The specific activity of radionuclides was determined by the following formula:(1)$$A=\frac{10^{6}}{k}\cdot \frac{S}{t\cdot m},$$
where ***A*** is the specific activity of the radionuclide [Bq/kg], ***k*** is the estimated number of recorded gamma rays per 10^6^ decays, ***S*** is the number of events in the total absorption peak minus the contribution of the detector background, ***t*** is the time [s], and ***m*** is the mass of the sample [kg]. The ***S*** value was determined as the difference between the number of events in the peaks in the spectrum with the sample and in the background spectrum normalized to the time of measurements with the sample, taking into account the absorption of gamma rays in the sample.

To calculate the activity of Th-232, U-235, and U-238, peaks from the gamma lines 238.6 keV (Pb-212), 143.8 keV (U-235), and 351.9 keV (Pb-214), respectively, were used. It was assumed that the secular equilibrium in the decay chain was not disturbed.

To check the reliability of the measurements, we performed calibration measurements with a sample of known activity. The activity of the K-40 isotope was measured in a sample of potassium dichromate (K_2_Cr_2_O_7_) weighing 297 ± 1 g in the same container (as the titanium samples). Using reference data on the chemical composition, the mass of the sample, the potassium isotopic composition, the half-life of K-40, and the yield of gamma quanta for decay, the activity of K-40 in the chromium sample was determined to be 1284 ± 4 Bq. The measured activity (taking into account the efficiency of recording gamma rays from the sample in a given container configuration, determined using the MCC program) was 1356 ± 43 Bq.

#### 2.2.3. Data Processing Technique and Preliminary Measurement Results

Measurements of the sum of the sample activity and the detector’s background were carried out for about 500 h for each sample. The measured spectra were compared with the detector background spectrum without samples. The contribution due to the sample was determined from the difference in the spectra. The sum-spectra show peaks at energies of 1460.8 keV, 1173.5/1332.5 keV, and 351.9 keV corresponding to the K-40, Co-60, and Pb-214/U-238 isotopes, respectively. In addition, two lines at 889.3 and 1120.6 keV were observed which indicate the presence of a short-lived isotope Sc-46 in titanium, with a half-life of 83.8 days. The presence of this isotope in a titanium sample is due to its production in titanium under the influence of cosmic ray neutrons [9].

## 3. Results and Discussion

### 3.1. Analysis of Radioactive Impurities in Master Alloys Used in the Production of Titanium Alloys Using ICP-MS

The introduction of master alloys into titanium alloys to correct its physical and mechanical properties is a standard procedure in the production of titanium alloys. Master alloys are introduced into the titanium sponge mixture before being pressed into consumable electrodes for the vacuum arc remelting (VAR) process. In order to study whether master alloys were a source of contamination with uranium and thorium, residual concentrations of uranium and thorium for master alloys used in titanium production at PJSC VSMPO-AVISMA Corporation (Verkhnya Salda, Russia) were selected and analyzed: chopped aluminum wire rod (Al-01), vanadium–aluminum alloy (VnAl-70V30Al), and aluminum–tin alloy (Al-Sn—50Al50Sn). These master alloys were introduced into the charge of titanium ingots in an amount averaging 4–5%, assuming that there were no secondary charge materials in the charge.

In addition to the studied master alloys, there are complex ones, consisting of 4–5 elements. Their amount in the charge of titanium-ingots can reach 15–18%.

The analysis of the chemical purity and content of residual U and Th in the samples of master alloys used in the production of titanium alloys (Table 3) showed that the most chemically pure material is the Al-Sn alloy, produced by VSMPO-AVISMA and obtained by liquid rolling. The lowest content of residual uranium and thorium was determined in the VnAl alloy, obtained by aluminothermy (the reduction of vanadium oxide with aluminum powder) at Uralredmet JSC (Verkhnyaya Pyshma, Russia).

However, the presence of U and Th in the master alloys reached such a level that if they were added to the electrode for the VAR and evenly distributed throughout the volume of the ingot, the requirements for radioactive purity would not have been met. It is possible to conclude that it is necessary to refrain from introducing any master alloys into the consumable electrode when producing ultra-low-background titanium alloys. The only alloy used for the manufacture of structural ultra-low-background titanium alloy is VT1-00. This is a titanium alloy, according to the OST 1 90013-81, and is an analog of a Grade 1 Ti alloy, according to ASTM B348.

### 3.2. Analysis of Radioactive Impurities in Titanium Metal Samples at Various Stages of Production Using ICP-MS

Samples of materials from various stages of VT1-00 titanium alloy production were selected to prove the possibility of producing structural titanium alloys with activity levels below 1 mBq/kg required for low-background experiments. Samples of titanium sponge were selected from the bulk of the TG-90-grade material (a mean of hardness 90 Brinell units, Russian standard—GOST 17746-96 [10]) manufactured in a bimetallic retort with reduced content of Cr, Ni, and Fe impurities (marked as TG-90 EK-3). A fraction with a particle size from 12 mm to 70 mm was selected for the manufacturing and further analysis of titanium ingots.

Titanium ingots of the VT1-00 alloy with a diameter of 180 mm and a weight of 15 kg were manufactured from the prepared samples of titanium sponge weighing 17 kg. The ingots were produced using three continuous stages of VAR. After the first and second remelting of the ingot, the ingot’s surface was washed. The ingot surface was machined after the third remelting. The top and bottom parts were removed from the final remelting ingots.

The production of the titanium sheet samples with dimensions of 350 × 500 × 2 mm was carried out by cold-rolling the ingots.

The analysis of residual U and Th in the samples taken from various stages of titanium metal production (Table 4) showed that, in a number of cases, it is possible to obtain titanium sponge with U equal to 1.30 ± 0.01 ppb and Th equal to 0.49 ± 0.10 ppb. Analyses of the titanium ingot and sheet showed that there was little change in the uranium and thorium. The main part of uranium and thorium was introduced during the production of ingots from titanium sponge TG-90EK-3.

Thus, it can be argued that if the primary material has ultra-low-background properties, contamination can be avoided during the subsequent technological stages of the production of structural titanium alloys. However, general patterns and quantitative assessments were not identified due to the insufficient number of samples subjected to measurements. This study plans to continue because the establishment of such a pattern will ensure the preservation of original ultra-low-background properties from the state of sponge titanium to rolled titanium.

### 3.3. Analysis of Radioactive Impurities in Titanium Metal Samples at Various Stages of Production Using the HP-Ge Method

To subtract the background of the detector from the measured data, the amount of gamma quanta in the total absorption peaks was normalized for 500 h of measurements. The normalized data are shown in Table 5.

The activity of radionuclides in titanium samples was obtained based on the sizes of the total absorption peaks of gamma rays in the measured spectra (Table 6). The measured numbers were corrected according to previously measured detector efficiency, simulated self-absorption in the sample, and gamma ray scattering associated with the geometry and physical properties of the detector sample.

## 4. Conclusions

Results on the impurity analysis of master alloys showed that their use greatly pollutes future titanium in terms of background radiation. An alternative is to avoid master alloys and use an ultra-low-background titanium sponge. Analysis of the results for uranium and thorium using ICP-MS and HPGe showed that the values obtained by the germanium detector are slightly lower, which can be explained by the homogeneity and larger mass of the samples. This may also be an indication of a violation of the secular equilibrium, which requires additional research.

Thus, the experiments carried out confirmed that it is possible to obtain ultra-low-background titanium of the VT1-00 grade in a standard technological process, provided that the titanium sponge is initially ultra-low-background.

## Figures and Tables

**Figure 1 materials-17-00832-f001:**
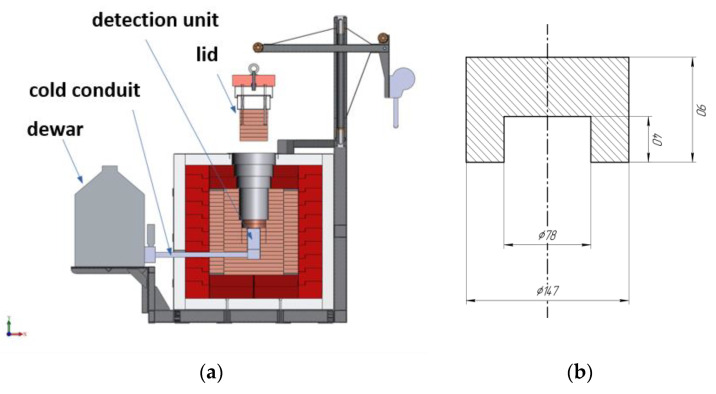
Scheme of a low-background gamma spectrometer in passive protection (**a**) and a sample scheme (mm) (**b**) for HP-Ge measurements made from a titanium ingot. The copper protection layer is highlighted in pink-brown, the lead layer is highlighted in red, and borated polyethylene is highlighted in white.

**Table 1 materials-17-00832-t001:** The operating mode of the NexION300D instrument for conducting impurity analysis of samples.

Nebulizer Type	Concentric (Meinhard), PFA
Spray chamber	Scott double-pass chamber, PFA
Argon flow rate, L/min	1.0
through the nebulizer	0.96
Plasma forming	15
Auxiliary	1.2
Generator power, W	1450
Potential on the analog segment of the detector, V	−1750
Potential on the pulse segment of the detector, V	1100
Detector discrimination threshold	12
Potential at the deflector, V	−10.5
Potential on quadrupole rods, V	−12
Sweeps/reading	30
Readings/replicate	10
Replicates	3
Dwell Times	100 ms for ^232^Th and ^238^U

The determination limit of U and Th was 0.01 ppb.

**Table 2 materials-17-00832-t002:** Calculated number of detected γ-quanta from a titanium sample per 10^6^ isotope decays (in the case of a decay chain—per one decay of the initial isotope of the chain) based on the simulation results.

Sample	Gamma Line, keV (Isotope/Decay Chain)								
	1332.5 (Co-60)	1460.8 (K-40)	186.2 (Ra-226/ U-238)	351.9 (Pb-214/ U-238)	185.7 (U-235/ U-235)	143.8 (U-235/ U-235)	911.2 (Ac-228/ Th-232)	238.6 (Pb-212/ Th-232)	661.7 (Cs-137)
	Estimated Number of Detected Gamma Rays per 10^6^ Decays								
Ti sponge	10,002 ± 100	974 ± 31	1674 ± 45	11,340 ± 106	28,028 ± 167	5560 ± 76	3571 ± 60	18,121 ± 135	14,915 ± 122
Ti ingot	3992 ± 63	411 ± 20	482 ± 24	3441 ± 59	7088 ± 84	1236 ± 36	1272 ± 36	5099 ± 72	5211 ± 72
Ti sheet	5165 ± 72	531 ± 23	669 ± 29	4839 ± 70	10,391 ± 102	1839 ± 44	1735 ± 42	7129 ± 84	7009 ± 84

**Table 3 materials-17-00832-t003:** Results on the analysis of the purity and content of residual uranium and thorium in alloy samples and their impact on the production of titanium alloys.

	Sample	Total Purity	Th	U
		wt.%	ppb	ppb
1	Al-01	99.70	64 ± 4	870 ± 60
2	VAl-70V30Al	99.82	0.24 ± 0.03	46.8 ± 0.3
3	Al-Sn-02	99.97	69 ± 2	335 ± 9

**Table 4 materials-17-00832-t004:** Results on the analysis of the purity and content of residual uranium and thorium in preparations from samples taken from various stages of titanium metal production.

No.	Samples	Th		U	
		ppb	Bq/kg	ppb	Bq/kg
1	Ti sponge	0.49 ± 0.10	0.002 ± 0.0004	1.30 ± 0.01	0.016 ± 0.001
2	Ti ingot	4.18 ± 0.06	0.017 ± 0.0002	2.67 ± 0.01	0.033 ± 0.001
3	Ti sheet	3.31 ± 0.12	0.013 ± 0.0005	2.45 ± 0.15	0.030 ± 0.002

**Table 5 materials-17-00832-t005:** The measured number of gamma quanta in total absorption peaks for samples made of titanium TG-90-EK-3 normalized for 500 h of measurements.

Samples	Gamma Line, keV (Isotope/Decay Chain)								
	1332.5 (Co-60)	1460.8 (K-40)	186.2 (Ra-226/ U-238)	351.9 (Pb-214/ U-238)	**185.7** (U-235/ U-235)	**143.8** (U-235/ U-235)	**911.2** (Ac-228/ Th-232)	**238.6** (Pb-212/ Th-232)	**661.7** **(Cs-137)**
	Estimated Number of Detected Gamma Rays per 10^6^ Decays								
Background	86 ± 6	61 ± 6	≤99	36 ± 6	≤99	26 ± 8	10 ± 3	70 ± 8	≤11
Ti sponge	62 ± 6	72 ± 6	≤99	52 ± 6	≤99	18 ± 7	15 ± 4	79 ± 8	54 ± 6
Ti ingot	36 ± 6	58 ± 8	≤153	38 ± 9	≤153	32 ± 15	≤10	77 ± 13	≤12
Ti sheet	60 ± 7	38 ± 6	≤165	130 ± 12	≤165	≤23	24 ± 6	159 ± 14	≤9

**Table 6 materials-17-00832-t006:** Calculated activity of radionuclides peaks for samples made of titanium TG-90-EK-3 [Bq/kg].

Samples	K-40	Co-60	U-238	U-235	Th-232	Cs-137
Ti sponge	≤0.08	≤0.004	0.005 ± 0.002	≤0.06	0.003 ± 0.002	0.010 ± 0.001
Ti sheet	≤0.008	≤0.001	0.0044 ± 0.0004	≤0.002	0.0036 ± 0.0003	≤0.0002
Ti ingot	≤0.01	≤0.0004	0.0008 ± 0.0003	≤0.002	0.0012 ± 0.0003	≤0.0003

## Data Availability

Data are contained within the article.

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
