# Peer review of "The Study of Radioactive Contaminations within the Production Processes of Metal Titanium for Low-Background Experiments"

_materials, 2024, doi:10.3390/ma17040832_

Round 1

Reviewer 1 Report

Comments and Suggestions for Authors

General remark: I am not sure if materials with contamination level of about 1 mBq/kg can be called ultra-pure (or with ultra-low levels of radioactive impurities). There are materials, like OFHC coper, or electroformed copper with proven radio-purity level below 10 micro_Bq /kg for Ra-226 or below 0.1 ppt for U/Th.

- Abstract: „It has been found out that all studied samples will be a source of uranium and thorium contamination of the final titanium-alloys” – not clear sentence.

- Abstract: “It has been established that the only product allowed obtaining ultra-low-background titanium was the VT1-00 which is manufactured without master alloys addition” – does it mean that the master alloys addition is the source of U/Th? One could write it clearly.

- Line 34: two dots at the end of the sentence.

- Ref. 1 and Ref. 2 seem to be a bit outdated.

- Ref. 3: cited paper is not related directly to neutrinos

- Line 50: Actually the “old” approach still works, but one needs to test large samples form the production, which will be used in the experiment after verification. This way the e.g. GERDA cryostat was build: Nuclear Instruments and Methods in Physics Research A 593 (2008) 448–453

- What are the parameters of the used ICP-MS and HPGe instruments (e.g. the sensitivity of HPGe for the applied geometries)?

- Table 1 (and Table 4): more clear would be to give directly efficiencies

- I understand that in the calculation of efficiencies, decays of respective isotopes were simulated, which include the emission probabilities. Maybe on can state this clearly.

- How the simulation of efficiencies were verified?

- Line 201: TG-90 with the index “EK-3”, fraction +12-70 mm – could you explain the nomenclature?

- Table 3: The results show a clear increase in U/Th going from form the sponge to the ingot. The final purity of the material (sheet) is e.g. for U at the level of 30 mBq/kg, which is way above the contamination of typical stainless steel (single mBq/kg). A comment here would be useful.

- Were the samples, for which results were reported in Tab. 4 and 5, the same?

- In table 5 instead of U-238 rather Ra-226 should be reported (The same for Th-232: Th-228 or Ra-228 should be quoted). If the samples investigated by ICP-MS and HPGe spectrometry were the same this would mean the radioactive equilibrium was broken e.g. in the U-chain.

- U/Th reported in Table 5 for the sheet are at the few mBq/kg level (4.4 mBa/kg for U/238/Ra-226) thus, clearly above 1 mBq/kg and comparable to stainless steel. A comment here would be useful what would be the advantage of Ti as structural material in low-background experiments.

Comments on the Quality of English Language

In general English is fine.  If possible a native speaker could read it. 

Author Response

Dear Reviewer

Thanks for the fruitful comments. We tried to answer for all questions and believe that it will enhance the article/

Q1.

General remark: I am not sure if materials with contamination level of about 1 mBq/kg can be called ultra-pure (or with ultra-low levels of radioactive impurities). There are materials, like OFHC coper, or electroformed copper with proven radio-purity level below 10 micro_Bq /kg for Ra-226 or below 0.1 ppt for U/Th.

Reply 1

 You are right that there is no commonly agreed term describing the material developed within the “low radioactive technologies and efforts” developing for ultra low radioactive detectors. We follow the idea that mBq activity is equal to ppb concentration. But ppb concentrations of impurities are usually refereed to when ultra pure chemical substances are discussed. At the same time we follow the practice accepted in the literature focused on these topics and our previous publication

Q2.

Abstract: „It has been found out that all studied samples will be a source of uranium and thorium contamination of the final titanium-alloys” – not clear sentence.

Reply 2

We added “ …all doped studied samples…”

Q.3.

Abstract: “It has been established that the only product allowed obtaining ultra-low-background titanium was the VT1-00 which is manufactured without master alloys addition” – does it mean that the master alloys addition is the source of U/Th? One could write it clearly.

Reply 3

Yes. You are right. The master alloys in titanium production are the sources of U/Th contamination. We changed the sentence in the Abstract.

Q4. 

Line 34: two dots at the end of the sentence.

Reply 4

Corrected

Q5.

Ref. 1 and Ref. 2 seem to be a bit outdated.

Reply 5

Corrected. Updated.

Q6.

Ref. 3: cited paper is not related directly to neutrinos

Reply 6.

Corrected. Updated.

Q7.

Line 50: Actually the “old” approach still works, but one needs to test large samples form the production, which will be used in the experiment after verification. This way the e.g. GERDA cryostat was build: Nuclear Instruments and Methods in Physics Research A 593 (2008) 448–453

Reply 7

Yes, the “old” approach works concerning structural alloys when the mass of the cryostat is less or equal to the single batch of the row material manufactured within the single industrial process. When the masses of the vessels exceed tens of tons, one  you have to take material from different batches and order with a large supply. Thus, the approach we propose is more efficient, reliable and predictable.

Q8. 

What are the parameters of the used ICP-MS and HPGe instruments (e.g. the sensitivity of HPGe for the applied geometries)?

Reply 8.

The relative efficiency of the HPGe for the applied geometries of the samples is 20%. 

We added the information in Table 1 for ICP-MS parameters.

Q9.

Table 1 (and Table 4): more clear would be to give directly efficiencies

Reply 9.

Yes, this is possible, but we believe that it would be more correct to provide direct modeling data and corresponding measurements.

Q10.

I understand that in the calculation of efficiencies, decays of respective isotopes were simulated, which include the emission probabilities. Maybe on can state this clearly.

Reply 10

We added the details.

“ The number of events at the peak of total absorption of gamma rays was determined from the model spectra. It was taken into account the branching factors in decay chains and the quanta emission per decay of nuclide.  The results of the calculation based on the simulation are presented in the Table 2.”

Q11.

How the simulation of efficiencies were verified?

Reply 11.

We added the details of the verification procedure.

“To check the reliability of the measurements, we performed a calibration measurement with a sample of known activity. The activity of the K-40 isotope was measured in a sample of potassium dichromate (K2Cr2O7) weighing 297±1 g in the same container (as the titanium samples). Using reference data on the chemical composition, mass of the sample, potassium isotopic composition, half-life of K-40 and the yield of gamma quanta for decay, the activity of K-40 in the chromium sample was determined to be 1284 ± 4 Bq. The measured activity (taking into account the efficiency of recording gamma rays from the sample in a given container configuration, determined using the MCC program) was 1356 ± 43 Bq.”

Q12.

Line 201: TG-90 with the index “EK-3”, fraction +12-70 mm – could you explain the nomenclature?

Reply 12.

We added the explanation.

Samples of titanium sponge were selected from the bulk of the TG-90 grade material (meaning hardness 90 in Brinell units, Russian standard - GOST 17746-96)  manufactured in a bimetallic retort with the reduced  content of Cr, Ni, Fe impurities (marked as  TG-90 EK-3). A fraction with a particle size from 12 mm to 70 mm was selected for titanium ingots manufacturing and further analysis.

Q13.

Table 3: The results show a clear increase in U/Th going from form the sponge to the ingot. The final purity of the material (sheet) is e.g. for U at the level of 30 mBq/kg, which is way above the contamination of typical stainless steel (single mBq/kg). A comment here would be useful.

Reply 13.,

We studied the process of radioactive impurities migration in the Ti production cycle. To do this we used not the material with the lowest U/Th concentration because we need to measure the changing in values of U/Th concentrations from stage to stage. If we had the sample with zero U/Th we did not see any changing down in case.

In the case of stainless steel, it is necessary to select "clean" steel through a lot of analysis. Maybe for tones of steel sheets. If you're lucky, the number of tests will be several hundred. In the case of titanium, using the developed technology, we obtain a guaranteed “pure” product in the form of a sponge, and then sheets with a yield of more than 90%.

Q14.

Were the samples, for which results were reported in Tab. 4 and 5, the same?

Reply 14.

Yes. There were the same samples.

Q15. 

In table 5 instead of U-238 rather Ra-226 should be reported (The same for Th-232: Th-228 or Ra-228 should be quoted). If the samples investigated by ICP-MS and HPGe spectrometry were the same this would mean the radioactive equilibrium was broken e.g. in the U-chain.

Reply 15.

Yes, it is presented as you mentioned above. The number of gamma quanta in total absorption peaks for Ra-226 is reported. It is also shown that it belongs to U-238 decay chain. Ac-228 and Pb-212 are reported from Th-232 decay chain. Yes you are right that radioactive equilibrium could be  broken. We point it out in the conclusion as a source of inconsistency of the HPGe and ICP-MS results. 

Q16.

U/Th reported in Table 5 for the sheet are at the few mBq/kg level (4.4 mBa/kg for U/238/Ra-226) thus, clearly above 1 mBq/kg and comparable to stainless steel. A comment here would be useful what would be the advantage of Ti as structural material in low-background experiments.

Reply 16.

We studied the process of radioactive impurities migration in the Ti production cycle. To do this we used not the material with the lowest U/Th concentration because we need to measure the changing in values of U/Th concentrations from stage to stage. If we had the sample with zero U/Th we did not see any changing down in case.

Reviewer 2 Report

Comments and Suggestions for Authors

Author Response

Dear Reviewer.

Thanks for the fruitful comments. We tried to answer all the questions and believe that it made the article better for readers.

Q1.

Line 26- -MS)

Reply 1.

Corrected …inductively coupled plasma mass-spectroscopy (ICP-MS)

Q2.

Line 31- abbreviation please

Reply 2.

Deleted VAR in the Abstract.  Added the explanation in the main text

Q3.

Line 34- abbreviation VT1-00 please explanation

Reply 3.

Added explanation in the main text.  «VT1-00 is a titanium alloy according to the OST 1 90013-81, which is an analog of Ti alloy Grade 1 according to ASTM B348»,

Q4

Line 35- Key words- ICP-MSand

please add- radionuclides

Reply 4.

Added and corrected

Q5.

Line 49 Does the info has a literary source that the demand for materials is changing so drastically??

Reply 5.

Megaton scale detectors are referred in the ASPERA program since 2005 (Astroparticle Physics Roadmap Phase I  Status and Perspective  of Astroparticle Physics in Europe - https://www.roma1.infn.it/people/capone/FNS_II/ASPERA_Roadmap.pdf)

Also  100 ton-year exposure, such as argon-based ARGO and xenon-based DARWIN and next-generation neutrino detectors  (Hyper-Kamiokande, JUNO, DUNE)   which are under development and construction are reffered in the following r article   -  https://doi.org/10.48550/arXiv.2203.14979

Q6.

Counting time 500 hours =aproxim 21 day

How long time was measured background for a sample,

Reply 6

The detector background was measured for 1000 hours before starting the measurements. It has been stable throughout the year. (We minimized the volume of air near the detecting part of the HPGe; when there is no sample, copper is placed).

Q7

? was one HP-Ge detector for all samples

Reply 7

Two identical HPGe detectors installed in the same room in the underground laboratory were used.

? or a detector for a sample,

The relative efficiency of the HPGe for the applied geometries of the samples is 20%.

Q8

Calibration standard?

Reply 8

To check the reliability of the measurements, we performed a calibration measurement with a sample of known activity. The activity of the K-40 isotope was measured in a sample of potassium dichromate (K2Cr2O7) weighing 297±1 g in the same container (as the titanium samples). Using reference data on the chemical composition, mass of the sample, potassium isotopic composition, half-life of K-40 and the yield of gamma quanta for decay, the activity of K-40 in the chromium sample was determined to be 1284 ± 4 Bq. The measured activity (taking into account the efficiency of recording gamma rays from the sample in a given container configuration, determined using the MCC program) was 1356 ± 43 Bq

?

Q9

What is the ICP_MS sensitivity for Th and U/

What is mass of a sample for analysis?

Reply 9

Instrument sensitivity is U > 50 000 cps/1 ppb. Rather, here it is worth talking about the detection limit (DL) for Th and U is 0.01 ppb according to the developed methodology.

Ti-based samples for ICP-MS analysis were 1 g.

Q10

How was estimated specific activity?

Th, Bq/kg 0,002+/- 0.0004??? .0,002+/- 0,001

Bq/kg 0,017+/- 0,0002?

Reply  10.

We carried out a mathematical conversion from ppb units to Bq/kg.

Q11

-Ge

What is MDA minimum detectable activity for all estimated radionuclides?

What is mass of a sample for analysis?

Reply 11

In this particular case a twice the error in the amount of measured activity for each line in the report is approximately MDA.

The ingot weight was 5.01 kg, the assembled sheets weight was  3.16 kg, the sponge weight was 0.3 kg

Reviewer 3 Report

Comments and Suggestions for Authors

The manuscript under consideration explores the possibility and techniques for producing industrial structural titanium with low radioactive contamination. The study involved measuring samples throughout the industrial process. The results obtained are interesting, and the methods employed are suitable for assessing radioactive contaminations at the 1 mBq/kg level.

As noted by the authors (lines 221-222), the results represent an intermediate step because a full industrial process has not yet been identified, and work is ongoing. Nevertheless, the main findings of the paper, such as the removal of master alloys and the stability of U and Th contamination over the manufacturing stages (Table 3), are worthy of publication and are of interest to readers.

I support the publication of the manuscript in Materials. Before publication, however, I would like the authors to address the following comments:

1) lines 47-52 Large-size cryostats are primarily constructed using copper and stainless steel, with only a relatively small mass attributed to titanium structural parts. The claim that titanium structural parts will reach masses of tens of tons is surprising. Can you confirm this statement and provide references or examples to support it? Additionally, what are the expected masses of VT1-00 required for future experiments?"

2) l.154 and l.254-259 In the Conclusion of the paper, the authors note that ICP-MS measurements consistently yield higher results than HP-Ge measurements and suggest a potential violation of secular balance in the samples. It is advisable to introduce this topic earlier in Section 3 and provide a more detailed analysis. Quantify the differences between these measurements (rather than stating that HP-Ge measurements are 'slightly lower' as mentioned in line 247) and discuss the potential reasons for a break in secular balance.

3) l. 192-193 “The only alloy for the manufacture of structural ultra-low background titanium alloy is VT1-00.” Can you provide a source or reference to support the statement that VT1-00 is the exclusive alloy suitable for manufacturing ultra-low background titanium structures?

Comments on the Quality of English Language

There are a few style, English, and language checks that you should perform. The caption in Table 3 is too wide. In line 41, 'modern universe' should be changed to 'present universe' or simply 'universe.' Additionally, in some cases, you may need to replace the use of the present perfect tense (e.g., 'have been carried out') with the past tense.

Author Response

Dear Reviewer!

Thanks for the fruitful comments. We tried to answer all the questions and believe that it made the article better for readers.

Q1.

1) lines 47-52 Large-size cryostats are primarily constructed using copper and stainless steel, with only a relatively small mass attributed to titanium structural parts. The claim that titanium structural parts will reach masses of tens of tons is surprising. Can you confirm this statement and provide references or examples to support it? Additionally, what are the expected masses of VT1-00 required for future experiments?"

Reply 1.

Megaton scale detectors are referred in the ASPERA program since 2005 (Astroparticle Physics Roadmap Phase I  Status and Perspective  of Astroparticle Physics in Europe - https://www.roma1.infn.it/people/capone/FNS_II/ASPERA_Roadmap.pdf)

Also  100 ton-year exposure, such as argon-based ARGO and xenon-based DARWIN and next-generation neutrino detectors  (Hyper-Kamiokande, JUNO, DUNE)   which are under development and construction are reffered in the following r article   -  https://doi.org/10.48550/arXiv.2203.14979

For example, the current vessel for DS-20k detector  needs  more than ~20t of  Ti or ~30t of the stainless steel  (   DarkSide-20k: Next generation Direct Dark Matter searches with liquid Argon arXiv:2312.03597v1 [hep-ex]  https://doi.org/10.48550/arXiv.2312.03597

Q2.

2) l.154 and l.254-259 In the Conclusion of the paper, the authors note that ICP-MS measurements consistently yield higher results than HP-Ge measurements and suggest a potential violation of secular balance in the samples. It is advisable to introduce this topic earlier in Section 3 and provide a more detailed analysis. Quantify the differences between these measurements (rather than stating that HP-Ge measurements are 'slightly lower' as mentioned in line 247) and discuss the potential reasons for a break in secular balance.

Reply 2.

Thank you for your comment. We see a violation of the secular equilibrium, but obtaining quantitative results requires additional long-term measurements with HPGe  for at least another year. These works are planned as a continuation of the described experiments. A discussion of the nature of the disruption of secular equilibrium is beyond the scope of this article.

Q3.

 3) l. 192-193 “The only alloy for the manufacture of structural ultra-low background titanium alloy is VT1-00.” Can you provide a source or reference to support the statement that VT1-00 is the exclusive alloy suitable for manufacturing ultra-low background titanium structures?

Reply 3.

It is the result of our study described in the present article

Q4.

Comments on the Quality of English Language

There are a few style, English, and language checks that you should perform. The caption in Table 3 is too wide. In line 41, 'modern universe' should be changed to 'present universe' or simply 'universe.' Additionally, in some cases, you may need to replace the use of the present perfect tense (e.g., 'have been carried out') with the past tense

Reply 4.

Corrected